# Rigid versus flexible: The effect of destination revitalization policy on tourists' travel behavioral intention

Wen Qin[1,2¤☉], Weilong Li[2¤☉], Yanli Peng[2¤], Juan Su[2*¤]

1 Business School of Central South University, Changsha, Hunan, China, 2 School of Tourism and Urban Rural Planning, Jishou University, Zhangjiajie, China

☉ These authors contributed equally to this work.
¤ Current address: School of Tourism and Urban Rural Planning, Jishou University, Yongding District, Zhangjiajie, Hunan Province, China
* 156656299@qq.com

## Abstract

Destination revitalization policies have been considered functional or institutional ones that would stimulate the tourism markets. This paper redefines the concept of destination revitalization policies as social cues and how they influence the social judgments and intention to travel by tourists. Based on the stereotype content model and temporal construal theory, we differentiate between two kinds of revitalization policies: flexible and rigid, and form a theoretical model to describe the differentiated psychological mechanisms of these policies and boundary conditions. Evidence from seven studies, including one semi-structured interview study and six scenario-based experiments, shows that flexible revitalization policies generally elicit stronger travel behavioral intention than rigid policies. This is brought about by differentiated social judgments: the flexible policies lead to increased perceptions of warmth, and the rigid policies to increased perceptions of competence. In addition, the effects are conditional to temporal distance. The flexible policies are more effective in case of close-term travel choices and rigid policies in case of long-term planning. By placing the destination revitalization policy in a new perspective as destination communication, the given study contributes to the existing knowledge on destination marketing and tourism policy governing and presents practical outcomes of how the policy should be designed to respond to how tourists make decisions and to ensure the sustainability of the development of a destination.

## Introduction

Public policies are commonly understood as institutional instruments designed to regulate markets, allocate resources, and guide collective action [1,2]. The traditional formulation of policy interventions in tourism research involved addressing it based

**Data availability statement:** We have publicly uploaded the dataset related to this study to the OpenICPSR repository. All data files required to replicate the findings of this study are available via the following link: OpenICPSR database (https://www.openicpsr.org/openicpsr/) and DOI: https://doi.org/10.3886/E244396V1.

**Funding:** This study was supported by the National Natural Science Foundation of China (No. 72362018; 42261042), and Key Project of Research on Teaching Reform in Ordinary Higher Education Institutions in Hunan Province (HNJG-20230688). Specifically, National Natural Science Foundation of China (No. 72362018; 42261042) provided funding support for the overall research project, including resources necessary for study implementation and data analysis. The Key Project of Research on Teaching Reform in Ordinary Higher Education Institutions in Hunan Province, China (No. HNJG-20230688) supported academic exchange and dissemination related to this research. The funding institution had no role in study design, data collection and analysis, decision to publish, or preparation of the manuscript.

**Competing interests:** The authors declare that they have no known competing financial interests or per-sonal relationships that could have appeared to influence the work reported in this paper.

on its functional capacities [3,4], as is the case with infrastructural development, stimulation or regulation of the market. Destination revitalization policies, in particular, are frequently implemented when the destination is in crisis, troubled image, or increased competition, and its main focus lies on recovery of the tourism performance [5–7].

In practice, however, revitalization policies are rarely confined to administrative implementation. They are publicly publicized and marketed to external audiences, that is expressly targeted on tourists as a marker of safety, enhancement or fresh appeal. As an illustration, in 2025 the Thai government created the quality label of the brand name Trusted Thailand in reaction to the safety issues and depleting tourist trust. These practices imply that destination revitalization policies do not simply readjust institutional settings or economic incentives; but also serve as communicative practices in which destinations describe themselves to their potential tourists [8,9]. Yet how tourists interpret these policy messages remains poorly understood.

Available studies contain valuable though partial information on destination revitalization policies. Tourism policy and destination governance research has drawn a major contribution of revitalization efforts of economic [10] or institutional perspectives [11], including market recovery, investment attraction, or governance efficiency as an outcome [12]. Although these studies explain what lays policies are meant to fulfil, they are weak in highlighting the way policies are understood by tourists as external audiences.

Conversely, studies of tourist perception and destination image emphasize the role of information cues in influencing the judgments and behavioral intentions of the tourists [13,14]. Nevertheless, in these literatures, public policy is hardly considered as a focal stimulus. Collectively, these streams are loosely related, and do not provide much scope of illuminating how tourists decode policy messages regarding revitalization, and why divergent behavioral impacts can be produced by policy efforts that have a similar intent behind them.

This paper conceptualizes the policies regarding destination revitalization as rigid or flexible on the basis of already recognized differences in the context of the rule-based, enforcement-based approaches to this policy, and those focused on flexibility and adaptive coordination [15,16]. The tourist view of these policy design mechanisms is that they act as salient governance cues. To decode the understanding of such cues, we use the model of stereotype content [17]. This model assumes that assessments of social actors are organized around two basic scales perceived warmth (benevolence and trustworthiness) and perceived competence (capability and efficacy). We, therefore, hypothesize that the inflexible or soft policy construction is a vital input, which influences the warmth and competence based assessment of tourists to their destination.

Nonetheless, studies have indicated that the decision contexts among the tourists are inherent in their travel decisions [18]. The temporal construal theory [19] suggests that people perceive and give careful consideration to different information about a situation, contingent on their time horizon. The distant-future choice is made according to abstract, high-level construals which focus on long-term orientation, strategic goals in general and the stability of the institutions (e.g., in the general sense that a

destination is managed in a predictable and trusted manner), whereas those of the near-future choices are based on concrete, low-level considerations with reference to immediate feasibility and convenience (e.g., whether current actions will enhance on-site experiences, alleviate immediate travel frictions or bring impressive benefits to visitors). Consequently, it becomes necessary to add the temporal distance as an essential circumstance that conditions the processing and evaluation of policy cues to explain, in detail, the functioning of the policies on tourists.

It is by combining these theoretical lenses, we come up with a theoretical model that we can then use to systematically review the ways in which the design of destination revitalization policies affect travel intentions via individual social-cognitive processes (perceived warmth and competence) and in which each of these processes is more dominant relative to the other, depending on the temporal perspective of the traveler. This study makes several contributions to the literature. Conceptually, it conceptually redefines destination revitalization policies as either functional or institutional interventions and relates them to socially meaningful cues that tend to influence tourist judgement of destinations. Theoretically, it is a combination of social judgment and temporal construal view on the issue of discovering when and how policy styles affect travel behavioral intention. In practice, the results can form a foundation of the development of revitalization policies with communicative characteristics that are relevant to tourists in the context of choice.

## Theoretical background and hypotheses development

### The stereotype content model

The stereotype content model (SCM), developed within social psychology, provides a foundational framework for systematically understanding the content and structure of social stereotypes [17]. It answers the basic question of how people create mentally the perceptions of social groups or collective actors. According to the model, the perceptions are arranged efficiently in two universal dimensions which include perceived warmth and perceived competence [20]. Warmth includes considerations of the intentions on the part of the actor (e.g., trustworthiness, friendliness), whereas competence constitutes considerations of the capacity of the actor to carry out the intentions (e.g., capability, efficacy) [21]. These two dimensions are not descriptive but are predictive in nature in that they collectively influence the following emotional and behavioral reactions towards the perceived target. The SCM has been fruitfully generalized to explain, however, interpersonal judgment as a result of the simplicity of the model and its explanatory strength, has been generalized to other fields, a consumer behavior being one of them [22,23].

The SCM has also been used in an increasing number of tourism studies to assess the way tourists consider countries and destinations as social units. Empirical data indicate that warmth- and competence-based judgments are used by tourists in creating destination images [23], and that the two dimensions are highly predictive of attitudes and visit intentions [24]. Nonetheless, the already available tools have been mostly concerned with comparatively stable destination images that have been formed due to culture or generated reputation [25,26], and little attention has been given to the reaction of tourists in relation to interventions undertaken by policy. The destination revitalization policies are salient publicly announced signings, especially in uncertain situations or in case of being seen as risky. Instead of being considered based on their functional efficacy, these policies give governance-related indications on which a tourist base his/her interpretation on the intentions of a destination to visitors and its abilities to provide safety and quality. The implementation of the SCM in this setting thus plays a requisite theoretical role: it specifies how policy features of revitalization policies are mentally carried over into the perceptions of warmness and competency hence bridging between policy design at the macro-level with tourist judgments and travel behavioral intentions at the micro-level.

### The temporal construal theory

The theory at the construal level serves as a source of interpretation of this phenomenon in that the higher the psychological distance of an event (temporal, social, spatial distance, etc.), the more abstract and high-level representations of

an event or object individuals use that are focused on its most important (goal-relevant) points of it. Alternatively, smaller psychological distance results in higher levels of concreteness with low-level construal mechanisms focusing on specific and contextual information [27]. One of the most basic and most studied dimensions of psychological distance is temporal distance. It is based on this that the theory of temporal construal was developed with a specific reason of mapping how the time horizon systematically changes the cognitive processing [28,29]. The theory is that when making decisions about far future, people put the most emphasis on abstract, high level qualities (e.g., the long established image of a destination as being trustworthy, stable in an institution, having symbolic value to the person making the decision). When it comes to almost immediate decisions, the focus is on tangible and low-level feasibility issues (such as immediate promotional discounts, improved quality of services or convenient booking) [19]. As far as destination policy evaluation is concerned, it means that the salience and persuasiveness of various policy cues remain dependent upon the planning horizon of the traveler, as either the abstract or concrete capability cues.

The past research revealed that concrete information is sought more by the tourists according to the temporal distance to the trip, and thus it is predicted that concrete information becomes more important in the decision-making of tourists at the near temporal distance [30,31]. Conversely, tourists who intend to visit a destination in the temporal distant tend to have a positive attitude towards destination promotion messages when employing high language abstraction [32]. The existing literature related to the temporal construal theory has generated new understanding of cognition and behaviors such as prediction, evaluation, and decision making in the fields of psychology and consumer behavior [33]. What's more, prior research has also examined when the characteristics of things and the information conveyed match the construal level, it is more beneficial for people's choices and decisions [34]. Based on this, this research will focus on temporal distance to explore the relationship between the revitalization policy and tourists' travel behavioral intentions under the context of tourism.

## Revitalization policy and travel behavioral intention

Destination revitalization policies represent a multidimensional policy approach aimed at restoring, sustaining, or enhancing the competitiveness and appeal of tourism destinations facing decline, disruption, or reputational crises [35]. In contrast to traditional destination development strategies, most of the revitalisation policies have been rather reactive but, at the same time, strategic responses to certain issues like over-tourism or market stagnation, damage to the environment, or governance failures [9,36]. These policies can be in the form of regulatory restructuring, public-private joints, focal investments, or socio-cultural projects which rebrand the destination into a competitive tourism environment. It is becoming more commonly considered that destination revitalization is not merely an economic necessity, but is a kind of adaptive governance, whereby destination authorities experiment with their institution to reestablish visitor confidence, quality standards and alignment between tourism and the sustainable development agenda [37,38].

Since it is a part of the broader context of destination revitalization, it is possible to distinguish policy strategies in terms of their level of enforcement, flexibility, and target orientation [15,16]. Particularly, there is an emphasis in strict revitalization policies that are defined as rule-based, top-down governance interventions that place an emphasis on order, discipline, and regulatory control [39,40]. Such policies are usually directed to tourist businesses, traders and local operators and not necessarily to the tourists themselves, as a courtesy to correct any market malpractice and confidence among people. As an example, the campaign of iron-fist tourism governance which was initiated in Zhangjiajie, China is an inflexible attitude in which the local government installed stringent control measures against forced trading, overpricing, and unlawful activities as a means to establish a secure, dependable and transparent tourism space. These policies are indicative of institutional determination and help create institutional long term destination credibility in terms of deterring opportunistic practices in the tourism system.

Flexible revitalization policies on the other hand will have an incentive oriented, participatory approach with the aim of improving the experiences of the tourists and increasing the demand [41]. These comprise/ consist of financial subsidies

and promotion and discounts of tickets and quality of services and cultural participation. Compared to the rigid ones, flexible policies can specifically address needs, preferences, and feelings of tourists and can even frame a destination as a place that is friendly, hospitable, and oriented towards its visitors [42]. It has already been emphasized by previous research studies how research-supported such strategies are effective in restoring the intention to travel, notably, during post-crisis situations, when they promote positive affective reactions and perceived reduced travel risks [43,44].

Empirical studies have revealed that service improvement and other demand-side policy instruments, including monetary incentives, promotion campaigns, and perceived cost reduction, can affect the behavioral intention of tourists and generate it in a significant positive way [31]. Remedial, incentive-based approaches tend to work best in post-crisis or recovery situations, whereby the destination insecurity and uncertainty among tourists are elevated [45]. Conversely, although the regulatory and enforcement-based policies can enhance the institutional purity of a destination, they are less prone to any immediate behavior change because of less flexibilities, slower to reflect, and poorer in communicating their value to the tourists [46]. Given this evidence, it is likely that the tongue will have a more direct and strong impact with the motivation of the tourists to travel with the implementation of flexible policies of revitalization. Accordingly, we propose the following hypothesis:

H1: Flexible revitalization policy will trigger stronger travel behavioral intention compared to rigid revitalization policy.

## The mediation effect of social judgments

The considerations of the destination by the tourists are not just determined by the tourism products and services but also about the perception of the manner in which a destination is governed [47]. In the stereotype content model, governance practices are salient social signals, which trigger warmth- and competence-based evaluations of collective actors. In line with this, one can anticipate that various revitalization policy styles will result in the creation of various social judgments, which subsequently influences the travel behavioral intentions of tourists.

Rigid revitalization policies lay emphasis on compliance of rules, standardization and efficiency of the process [15,48]. These qualities are indications of great administrative ability and coordination, which is associated with the competence dimension of social judgment. According to the previous research, regulation and efficiency-based governance generally reflects on professionalism and competence [49]. Perceived destination competence lessens uncertainty, boosts belief in reliability of the services to be received and boosts expectations of competently managed tourism experience, thus boosting the movement behavior intentions of the visitor in the tourism industry.

Compared to this, flexible revitalization policy emphasises flexibility, responsiveness as well as sensitivity to stakeholder needs [15,50]. These attributes are in line with the warmth dimension because they portray benevolent overtones coupled with people-focused governance orientation. Available literature indicates that the perceived warmth is also elevated with inclusive and responsive governance practices, which indicate care, approachability and trustworthiness in the practices [51,52]. Such warmth perceptions, in tourism decision-making, create the feeling of emotional attachment and belief towards the destination, which are instrumental factors in generating positive behavioral intentions of travel.

Combined, based on the stereotype content model, this experimental proposal hypothesizes that the styles of the revitalization policy affect the intentions of tourists to travel around in an indirect manner based on the dissimilar social judgment routes. In particular, the perceived competence can be boosted by strict policies, the perceived warmth can be boosted by flexible policies, and these perceptions are dominant mediating variables between designing a policy and the intention to travel. Accordingly, we hypothesize:

H2: Social judgement mediates the relationship between revitalization policy and travel behavioral intention.

H2a: Perceived warmth mediates the relationship between flexible revitalization policy and travel behavioral intention.

H2b: Perceived competence mediates the relationship between rigid revitalization policy and travel behavioral intention.

## The moderations effect of temporal distance

The study on travel decision-making has confirmed the central role of temporal factors, especially the planning horizon, in the acquisition, processing and evaluation of information among the tourists. The temporal construal theory is one theoretical lenses to be used to explain this influence [27,29]. It hypothesizes that cognitive construal is systematically influenced by psychological distance: a decision regarding the distant future is based on high-level, abstract characteristics (e.g., core values of a destination, institutional credibility, long-term reliability), whereas a decision regarding the near future depends on low-level, concrete characteristics (e.g., immediate convenience and situational feasibility) [19].

It follows within this point of view that the social judgment processes that are elicited by various policy styles are critically moderated by temporal distance. Interestingly, there is a difference in the nature of the policies of rigid and flexible revitalization in terms of the nature of their time orientation. Rigid policies are normally developed with the introduction of long term effects based on continued enforcement, standardization and continuity of the institution [48,53]. The characteristics are congruent with the distant-future decisions, in which the tourists form high-level construal and value enduring and long-lasting destination characteristics. Rigid policies in these situations serve as reckonings of governance and order in the processes [30,32], hence reinforcing their influence through the perceived competence route. The example of this is through the multi-year safety accreditation scheme of a destination; the scheme enhances trust in the managerial ability of the destination with regard to next year travel considerations.

On the other hand, the aim of the flexible policies is frequently to create a greater amount of short term, situational gains by focusing more on adaptability, responsiveness, and facilitation in the short term [15]. The closer the travel time, the closer to low construal tourist changes to be and to pay closer attention to the concrete experiential details they can. Limited-time offerings, flexible terms of booking, or responsive adjustments of the services are flexible policy instruments that directly adjust to these short-term needs [54]. By doing this they indicate concern and visitor orientation and intensify policy impacts in the warmth-based perceived pathway [30,52]. As an example, a discount that is given when returning to visit to a destination on a coming weekend can increase the perceptions of care and welcome of the destination. Accordingly, we hypothesize:

H3: Temporal distance moderates the mediation effect of social judgments between revitalization policy and travel behavioral intention.

H3a: When temporal distance is short, flexible revitalization policy will activate stronger travel behavioral intention through the mediation effect of perceived warmth.

H3b: When temporal distance is long, rigid revitalization policy will activate stronger travel behavioral intention through the mediation effect of perceived competence.

Temporal distance also primes the global preference of tourists in various policy styles using a construal fit mechanism [19,29]. According to previous studies, it is easy to make a decision when there is agreement between the rationales of an option and situational performance, coupled with the intensity of construal [34]. Such alignment helps people to have a higher cognitive fluency and evaluative consistency, leading to a heightened behavioral intention.

The flexible revitalization policies are generally context-sensitive, elaborate, and adaptive governance provisions [15,16]. They focus on short-term modulations, sensitivity to timely demands, and task-dependent modulations, which are more features related to the low-level construal [33]. Therefore, in cases where temporal distance is low and where the tourists are concerned with tangible viability and direct experience, the flexible policies are more inclined to appeal to the decision frame and evoke better intentions to travel behavior.

Rigid revitalization policies on the contrary are more abstract in nature and general and holistic [35]. They are based on regular rules, consistent structures, and structural institutional patterns and this feature corresponds with high-level construal. Tourists are prone to consider destinations in terms of their overarching structure, predictability, and reliability when making the decisions that relate to traveling in distant future. In this case, rigid policies will offer a superior construal fit and consequently resonate more in the generation of travel behavioral intentions. Accordingly, we propose that:

H4: Temporal distance moderates the relationship between destination revitalization policy and travel behavioral intention.

H4a: When temporal distance is short, tourists may have a stronger travel behavioral intention of flexible (vs. rigid) revitalization policy.

H4b: When temporal distance is long, tourists may have a stronger travel behavioral intention of rigid (vs. flexible) revitalization policy.

Building on the above literature, this study proposes a conceptual model that delineates the mediating and moderating mechanisms through which destination revitalization policies influence tourists' behavioral intention (see Fig 1). The proposed hypotheses were tested across seven studies (see Table 1). A pilot study using semi-structured interviews explored the impacts of rigid and flexible policies and provided initial insights into the mediating role of social judgments. Study 1a, 1b employed scenario-based experiments to establish the main effect. Study 2 and Study 3 further validated the mediating roles of perceived warmth and perceived competence. Study 4 and Study 5 extended the findings by examining the moderating effect of temporal distance, thereby enriching the theoretical and empirical contributions of the model.

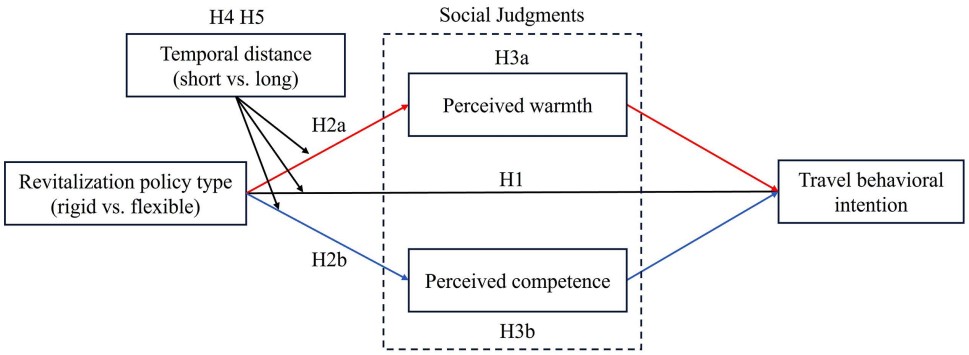

**Fig 1. Theoretical framework.** Source: Author's own work.

**Table 1. Outline of studies.**

| Hypotheses Test | Study | IV manipulation | DV operationalization | Samples |
|---|---|---|---|---|
| Main effect (H1) | Pilot study | Non | Non | 64 on-site tourists |
| | Study 1a (see Appendix in S1 File) | policy announcement (graphics) | Measurement of TBI | 100 online tourists |
| | Study 1b (see Appendix in S1 File) | policy news video (video) | Receive coupons or not | 87 on-site tourists |
| Mediate effect (H1-H2) | Study 2 | policy announcement (graphics) | Measurement of TBI | 270 online tourists |
| | Study 3 | policy news video (video) | Receive coupons or not | 265 on-site tourists |
| Moderated effect (H1-H3) | Study 4 | policy news video (video) | Measurement of TBI | 420 on-site tourists |
| | Study 5 (see Appendix in S1 File) | policy announcement (graphics) | Receive coupons or not | 500 online tourists |

Note: TBI = travel behavioral intention.

## Pilot study

The pilot study was meant to examine the perception of destination revitalization policies by the tourists and their perception in relation to social judgment and intentions to travel. Because of the exploratory type of the stage, qualitative design exploration using semi-structured interviews was chosen to produce first approximate empirical information, and shape the design of the following scenario-oriented experiments.

## Method

**Research procedure and samples.** The interviews were conducted between October 6 and October 11, 2023. Participants were recruited using a snowball sampling technique, which is commonly employed in exploratory tourism research to access individuals with relevant travel experience and to enhance participants' willingness to share detailed perceptions [55]. The first respondents were chosen on the basis of the professional and social networks of the researchers and on the node of the recommended respondents that will qualify as the study participants.

In order to achieve the relevance to the research objectives, the participants the following selection criteria were put: (1) they had already been to a destination and had at least one leisure trip in the last two years; (2) they could be conversant with specific destination revitalization policy-related information (e.g., safety and tourism recovery regulations, promotional policies) via media or through their own experience; (3) they were able to talk about their perceptions of destinations revitalization policies and related travel intentions.

There were 64 tourists that took part in the study. The interviews were carried out through telephone, and the rest of those were in person. Before the commencement of the interviews, study participants were told the purpose of research, contents of the interview and the free will to participate. All the respondents were informed verbally and provided their informed consent. In telephone interviews, the consent was audio-recorded, whereas in face-to-face interviews, it was provided orally on the presence of a research assistant.

The average duration took during the interviews was five to eight minutes and was recorded sound. The contents of all the recordings were transcribed word-to-word to be analyzed later. A description of the profiles of the interviewees is presented in Appendix E in S1 File.

**Research instruments.** In advance research, there were studies on tourism policy perception and social judgment and based on this, a semi-structured interview guide was created [31]. The interview protocol addressed three thematic areas aiming to capture the perceptions of the implementation approach of the destination revitalization policies; (1) the travel behavioral intentions; (2) social judgment about the various implementation approaches of the revitalization policies. Appendix B in S1 File reports the interview plan and the coding scheme.

For data analysis, a systematic coding procedure was applied. Following established qualitative research practices [56], two doctoral students independently and blindly coded interview transcripts. Based on classifications used in prior studies [57], statements referring to revitalization policies were coded as rigid (1) or flexible (0). Perceived warmth, perceived competence, and travel behavioral intention were coded dichotomously (1 = present; 0 = absent).

Inter-coder reliability exceeded 95% [58]. For the small number of inconsistent coding cases, three postgraduate students and one professor of tourism management jointly reviewed the transcripts and reached a consensus. Examples of the coding process are presented in Table 2.

## Findings

Among the 64 respondents, 36 participants referred primarily to rigid destination revitalization policies, while 28 participants referred to flexible policies. Of those discussing rigid policies, 97.22% expressed perceptions of competence, and

**Table 2. Coding examples of the original interview statements (Pilot study).**

| Construct | Coding | Interview statements (interviewee) |
|---|---|---|
| Destination revitalization policy | Rigid policy (1) | "Formation of law enforcement teams to eliminate vicious incidents in scenic spots." (Interviewee 12, female) |
| | Flexible policy (0) | "Free admission for visitors, hotel discounts, easy parking." (Interviewee 29, female) |
| Perceived competence | Perceived competence (1) | "This reflects local capacity and capability." (Interviewee 16, female) |
| | No perceived competence (0) | "This is the responsibility of every scenic government." (Interviewee 26, male) |
| Perceived warmth | Perceived warmth (1) | "The city is very welcoming and serves the people." (Interviewee 19, male) |
| | No perceived warmth (0) | "None of these appeals to me enough. I'm more interested in the landscape." (Interviewee 15, female) |
| Travel behavioral intention | Generate intention (1) | "If I ever get the chance, I'll definitely travel here." (Interviewee 21, male) |
| | No generate intention (0) | "Personally, I don't feel it would increase my willingness to travel." (Interviewee 09, female) |

2.78% expressed perceptions of warmth. In contrast, among respondents referring to flexible policies, 89.29% expressed perceptions of warmth, and 10.71% expressed perceptions of competence.

Chi-square test results indicated statistically significant differences between rigid and flexible policies with respect to perceived warmth ($\chi^2(1) = 41.94$, $p < 0.001$), perceived competence ($\chi^2(1) = 41.52$, $p < 0.001$), and travel behavioral intention ($\chi^2(1) = 4.53$, $p = 0.033$). These results are illustrated in Fig 2.

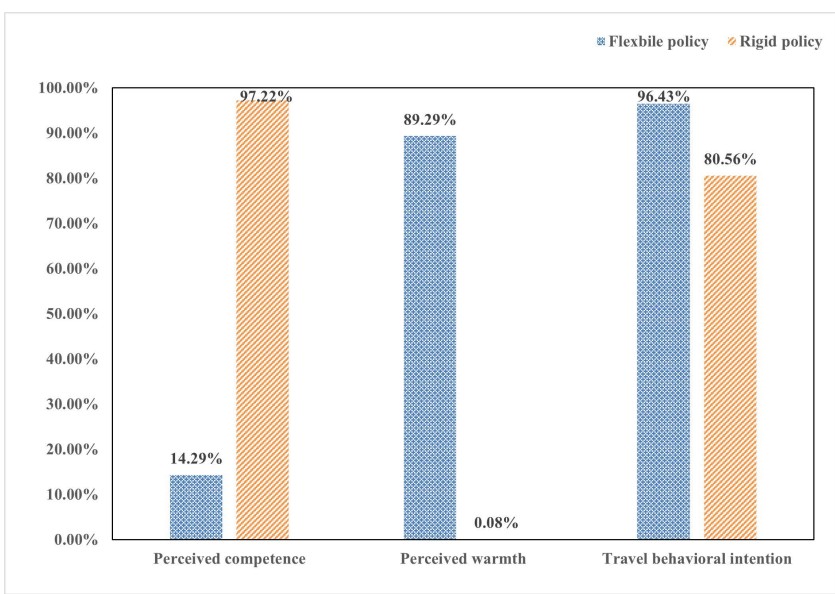

**Fig 2. The impact of implementation approach of tourism destinations' revitalization policy on tourists perceived warmth and perceived competence (pilot study).** Source: Author's own work.

## Discussion

The interview results indicate that tourists form different social judgments under different destination revitalization policies, providing preliminary support for H1 and H2. Specifically, flexible policies are more frequently associated with perceived warmth, whereas rigid policies are more strongly linked to perceived competence. Moreover, flexible revitalization policies evoke a higher level of travel behavioral intention (96.43%) than rigid policies (80.56%, see Fig 2).

## Study 2

Study 2 employed a scenario-based online experiment to test the main effects of destination revitalization policy style and the mediating roles of perceived warmth and perceived competence, as well as the moderating effect of temporal distance. A between-subjects experimental design (rigid vs. flexible) was adopted, in which participants were randomly assigned to one of the policy conditions.

### Research procedure and samples

Study 2 employed a scenario-based online experiment to examine the effects of destination revitalization policy type (rigid vs. flexible) on tourists' travel behavioral intentions and the mediating role of social judgments. Participants were recruited from December 9th, 2023 to January 9th, 2024 through Credamo.com, a widely used professional online survey platform in China that supports randomized assignment and participant screening [31].

Participation was entirely voluntary. Before accessing the questionnaire, all participants were provided with a detailed explanation of the research purpose, procedure, data usage, anonymity, and their right to withdraw at any time without penalty. Participants could only continue with the study after giving an informed consent. No other data that entailed personal privacy or information that can be identified as belonging to particular individuals were collected other than basic demographic data (e.g., age and gender).

The Ethics Committee of the School of Tourism and Urban Rural Planning, Jishou University revised and approved the study protocol, which included the participant recruitment, consent protocol, and experimental materials. Participation was done with informed consent which was obtained electronically.

There were 270 valid responses ($N_{rigid}$ = 135; $N_{flexible}$ = 135). The participants were selected randomly to take part in either of the two experimental conditions. The sample consisted of 67% female and 33% male respondents, with the largest age group being 26–35 years (44.4%). Detailed demographic information is reported in Table 2.

### Research instruments

The experimental stimuli described a fictitious destination ("City A") and a hypothetical traveler ("Xiaoli") to avoid confounding effects of prior destination familiarity or real-world experiences [59]. Participants were instructed to imagine that Xiaoli had learned about City A's destination revitalization policy through online information prior to making a travel decision.

Two versions of destination revitalization policy were designed to reflect rigid and flexible policy styles, respectively. Both versions shared the same structure, layout, and policy domains (tourism order, marketing order, and tourism safety) but differed systematically in policy tone and implementation approach.

In the rigid policy condition, the policy emphasized rule enforcement, standardization, and formal regulation (e.g., strengthening law enforcement teams, cracking down on illegal operations, and strictly regulating market order). In the flexible policy condition, the policy emphasized guidance, service facilitation, and adaptive support (e.g., establishing tourist service desks, expanding volunteer service teams, and promoting honest business practices and positive incentives). These materials were presented visually in the form of policy posters to enhance realism and consistency across conditions (see Appendix C in S1 File).

Participants were asked to read and imagine according to the scenarios and complete a four-part questionnaire. In order to ensure participants could accurately understand all scales, we used a formal backtranslation process [60]. Then they answered questions about scenario authenticity and evaluated scales of key variables (1 = strongly disagree, 7 = strongly agree). Questions about the scenario authenticity were adapted from Yi et al. [61], including "1. For me, there is no difficulty in understanding the given situation of the material" and "2. In real life such a scene could happen". For manipulation checks, participants were first informed of the definitions of rigid versus flexible policy and were asked to evaluate to what extent they agree with two statements: "I believe that the policies enacted by the city are rigid/flexible". Third, travel behavioral intention was measured by a 5-item scale (e.g., "I expect to travel to the destination in the future"; Cronbach's α = 0.90) adapted from Su et al. [62]. After measuring the independent variables, six items (adapted from Chang & Kim [63]) were used to measure the perceived warmth (I feel that destination A is "warmth", "friendly" and "kind"; Cronbach's α = 0.94) and perceived competence (I feel that destination A is "competence", "capable" and "efficient"; Cronbach's α = 0.94). Then, participants were asked to answer basic demographic questions. Finally, participants who completed the questionnaire received a small compensation through Credamo. Full measurement items are listed in Appendix D in S1 File.

**Results.** *Manipulation Check*. Independent-sample *t*-tests confirmed that participants clearly distinguished between the two policy conditions based on the experimental stimuli. Specifically, the rigid revitalization policy was perceived as significantly more rigid than the flexible policy ($M_{rigid}$ = 5.61 vs. $M_{flexible}$ = 2.08, $t$ = 21.17, $p < 0.001$), whereas the flexible revitalization policy was perceived as significantly more flexible than the rigid policy ($M_{flexible}$ = 6.20 vs. $M_{rigid}$ = 2.49, $t$ = −23.09, $p < 0.001$).

*Main Effects of Policy Type*. A one-way ANOVA was conducted to test Hypotheses H1–H3. The results revealed significant differences between policy types in perceived warmth ($F_{(1, 268)}$ = 152.70, $p < 0.001$, $\eta^2$ = 0.36), perceived competence ($F_{(1, 268)}$ = 63.02, $p < 0.001$, $\eta^2$ = 0.19), and travel behavioral intention ($F_{(1, 268)}$ = 7.80, $p < 0.01$, $\eta^2$ = 0.28). Compared with rigid policies, flexible policies elicited higher travel behavioral intention ($M_{flexible}$ = 5.80, $SD$ = 0.81; $M_{rigid}$ = 5.47, $SD$ = 1.10) and stronger perceived warmth ($M_{flexible}$ = 6.00, $SD$ = 0.77; $M_{rigid}$ = 4.03, $SD$ = 1.69). In contrast, rigid policies generated higher perceived competence than flexible policies ($M_{rigid}$ = 5.97, $SD$ = 0.88; $M_{flexible}$ = 4.60, $SD$ = 1.80). These results support H1 and provide preliminary evidence for H2a and H2b.

*Mediation Analysis*. To further examine the mediating effects, a mediation analysis was conducted using PROCESS Model 4 [64]. The independent variable of the policy type (rigid = 1, flexible = 0), perceived warmth and perceived competence as parallel mediators, and travel behavioral intention as the dependent variable were identified. The direct effect of policy type on travel behavioral intention was significant ($b$ = −0.33, $SE$ = 0.12, 95% $CI$ [−0.56, −0.11]). Both indirect effects were also significant: perceived warmth ($b$ = −0.29, $SE$ = 0.10, 95% $CI$ [−0.49, −0.11]) and perceived competence ($b$ = 0.17, $SE$ = 0.06, 95% $CI$ [0.05, 0.27]). These results suggest that the perceived warm and perceived competence mediate to some extent the relationship between the type of destination revitalization policy and travel behavioral intention which supports H2 (see Fig 3).

**Control variable and alternative explanation.** Given that traveler knowledge contributes to lessening the uncertainty in travel, and that consumer tastes and actions are determined by tourist knowledge [65], and since age-related variations can possibly influence consumer attitudes and behaviors [66], both age and tourist knowledge were incorporated as the control variables. Tourist knowledge was measured using a three-item scale adapted from Wong and Yeh [66] (Cronbach's α = 0.84).

After controlling for these variables, the influence of destination revitalization policy type on perceived warmth, perceived competence and travel behavioral intention were still significant (perceived warmth: $F_{(1, 266)}$ = 154.87, $p < 0.001$; perceived competence: $F_{(1, 266)}$ = 60.75, $p < 0.001$; travel behavioral intention: $F_{(1, 266)}$ = 5.89, $p = 0.02$). Further mediating the process analyses based on PROCESS Model 4 showed that neither tourist knowledge ($b$ = −0.07, $SE$ = 0.05, 95% $CI$ [−0.18, 0.03]) nor the age ($b$ = −0.01, $SE$ = 0.02, 95% $CI$ [−0.05, 0.04]) has a significant mediating effect in terms of the relationship between destination revitalization policy and travel behavioral intention.

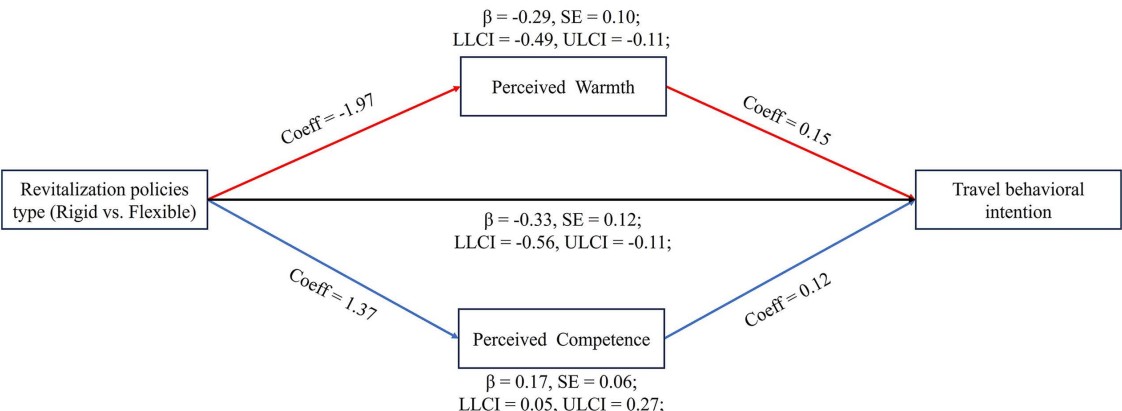

**Fig 3. Mediating effects of perceived warmth and perceived competence.** Source: Author's own work.

**Discussion.** Study 2 validated the main effects and showed that flexible (vs. rigid) policies enhanced perceived warmth, whereas rigid (vs. flexible) policies enhanced perceived competence, both of which partially mediated their respective effects on travel behavioral intention. The results supporting H1 and H2.

## Study 3

To enhance the external validity of the findings and address the limitations of prior scenario-based online experiments, Study 3 adopted a field-based experimental design using video stimuli and a behavioral outcome measure.

### Research procedure and samples

Data were collected on site in the core scenic area of Zhangjiajie from 09/01/2024 to 18/01/2024. A field intercept sampling technique was employed to recruit tourists who were actively visiting the destination. Potential participants were approached by trained research assistants in various times of the time of the day at predetermined points (e.g., at the entrances of the scenic areas, at rest spots and waiting zones). Screening of the participants was done on the following criteria: (1) age, i.e., they had to be at least 18 years; (2) visiting the destination in leisure; (3) the participant had to be in a position to watch a short video and answer the questionnaire on their own.

Before the respondents participated, they were made aware of the scholarly intention of the research, voluntariness of the proposed study and how they could voluntarily pull out anytime without any form of penalty. Data collection was started through informed consent which was informed verbally. The study protocol was reviewed and approved by the Ethics Committee of the School of Tourism and Urban Rural Planning, Jishou University. No personal privacy information was collected beyond necessary demographic variables.

Once the consent was obtained, the participants were assigned, randomly, to either one of two experimental conditions (rigid revitalization policy and flexible revitalization policy). They were exposed to the specified video stimulus on a tablet with headphones and then were asked to fill in a short questionnaire. In order to obtain the travel behavioral intention with the help of the behavioral indicator instead of self-report alone, the participants could be provided with a hotel accommodation voucher of RMB 50 and inquired whether they expect to accept it [67]. The behavioral outcome was the development of an acceptance of the coupon.

A total of 265 valid responses were received (53.6% female; 40.8% aged 36–45), with 129 participants assigned to the rigid-policy condition and 136 to the flexible-policy condition.

## Research instruments

Two video stimulus materials were constructed to depict the direct policies of revitalizing destinations which are rigid and flexible. The rigid-policy video rested on the put-principle-based tourism governance practices of Zhangjiajie which focus on strict control, uniform action and following of rules and regulations. The flexible-policy video was based on how Zibo governs its tourism services when it experienced a boom of barbecue tourism, which emphasized on friendliness, adaptability, and visitor-oriented services. The two videos were balanced against length, narrative form, presentation style and density of information in order to control extraneous variation (see Appendix C in S1 File).

After the video exposure, the participants filled out measurement scales in terms of perceived warmth, perceived competence, and control variables. The behavioral intention of travel was defined as a real choice behavior, which was represented by the uptake of hotel accommodation coupon.

The items of the questionnaires were formally back-translated into a different language to have linguistic equivalence and meaning. At the final of the questionnaire, demographic variables such as age and gender were gathered.

**Results.** *Manipulation Check*. An independent samples t-test tested the effectiveness of the policy manipulation. The subjects in the rigid policy condition felt that the policy is much more rigid compared to the individuals in the flexible policy condition ($M_{rigid} = 5.68$, $M_{flexible} = 2.31$; $t = 25.53$, $p < 0.001$) which indicated that there was successful differentiation between the rigid and flexible policy environment.

*Main Effects of Policy Type*. The results of one-way ANOVA showed significant difference among the participants in terms of social judgments among different policy conditions. Participants in the flexible policy condition reported significantly higher perceived warmth ($M = 5.33$, $SD = 1.48$) than those in the rigid policy condition ($M = 2.84$, $SD = 1.51$; $F(1, 268) = 152.70$, $p < 0.001$, $\eta^2 = 0.36$). Conversely, participants exposed to the rigid policy condition reported higher perceived competence ($M = 5.84$, $SD = 0.91$) than those exposed to the flexible policy condition ($M = 3.13$, $SD = 1.50$; $F(1, 268) = 63.02$, $p < 0.001$, $\eta^2 = 0.19$).

*Behavioral Outcome*. To assess travel behavioral intention using a behavioral indicator, a binary logistic regression was conducted with coupon acceptance as the dependent variable. Results showed that policy type significantly predicted participants' willingness to accept the coupon ($B = 0.69$, $Wald = 4.00$, $p = 0.046 < 0.05$). Specifically, participants in the flexible policy condition exhibited a higher acceptance rate (86.96%) than those in the rigid policy condition (80.31%), providing further support for H1.

*Mediation Analysis*. The mediating roles of perceived warmth and perceived competence were examined using PROCESS macro Model 4 with 5,000 bootstrap samples. The results indicated a significant direct effect of policy type on behavioral intention ($\beta = -0.49$, $SE = 0.12$; 95% CI [−0.72, −0.25]). In addition, both perceived warmth ($\beta = -0.29$, $SE = 0.10$; 95% CI [−0.50, −0.12]) and perceived competence ($\beta = -0.48$, $SE = 0.13$; 95% CI [−0.74, −0.24]) exhibited significant indirect effects. These findings support H2a and H2b (see Table 3).

**Control variable and alternative explanation.** Given the use of video-based stimuli, two media-related factors that may influence tourists' attitudes and behavioral responses were controlled: sense of presence and emotional arousal. Sense of presence has been shown to shape users' perceptions and behaviors in virtual environments and to enhance destination image and visit likelihood in destination marketing contexts [68]. It was measured using four items adapted from Verhagen et al. [69] (Cronbach's α = 0.82). Emotional arousal, which plays an important role in decision-making and is commonly elicited in tourism marketing communications, was measured using five items adapted from Jiang et al. [70] (Cronbach's α = 0.86).

When these variables were included as covariates, ANCOVA results indicated that the effects of policy type on perceived warmth ($F(1, 261) = 188.75$, $p < 0.001$, $\eta^2 = 0.42$), perceived competence ($F(1, 261) = 307.87$, $p < 0.001$, $\eta^2 = 0.54$), and travel behavioral intention ($F(1, 261) = 17.26$, $p < 0.001$, $\eta^2 = 0.06$) remained significant. Additional mediation analyses using PROCESS Model 4 further showed that neither sense of presence ($b = -0.0003$, $SE = 0.01$; 95% CI [−0.01, 0.03]) nor arousal ($b = -0.003$, $SE = 0.01$; 95% CI [−0.03, 0.02]) significantly mediated the relationship between destination revitalization policy and travel behavioral intention.

**Table 3. Mediation analysis results (Study 3).**

| | M1(PW) | | | | M2(PC) | | | | Y(TBI) | | | |
|---|---|---|---|---|---|---|---|---|---|---|---|---|
| | β | SE | LLCI | ULCI | β | SE | LLCI | ULCI | β | SE | LLCI | ULCI |
| Constant | 5.33 | 0.13 | 5.08 | 5.58 | 3.13 | 0.11 | 2.92 | 3.34 | 5.87 | 0.08 | 5.71 | 6.03 |
| X(RP) | −2.49 | 0.18 | −2.85 | −2.13 | 2.71 | 0.15 | 2.41 | 3.01 | −0.49 | 0.12 | −0.72 | −0.25 |
| $M_1$(PW) | – | – | – | – | – | – | – | – | 0.12 | 0.04 | 0.04 | 0.19 |
| $M_2$(PC) | – | – | – | – | – | – | – | – | −0.18 | 0.05 | −0.27 | −0.09 |
| | $R^2 = 0.41$, $F (1, 263) = 183.18$, $p < .001$ | | | | $R^2 = 0.54$, $F (1, 263) = 311.05$, $p < .001$ | | | | $R^2 = 0.06$, $F (1, 263) = 16.98$, $p < .001$ | | | |
| Indirect effect | | | | | | | | | Effect | SE | LLCI | ULCI |
| PR→PW→TI | | | | | | | | | −0.29 | 0.10 | −0.50 | −0.12 |
| PR→PC→TI | | | | | | | | | −0.48 | 0.13 | −0.74 | −0.24 |

Note: RP = revitalization policy; PW = perceived warmth; PC = perceived competence; TBI = travel behavioral intention; LLCI = lower limit of confident interval; ULCI = upper limit of confident interval.

**Discussion.** Building on the limitations of Study 2, Study 3 employed a field experiment with actual tourists to improve external and ecological validity. To mitigate the intention–behavior gap, a hotel accommodation coupon was used as a behavioral proxy for travel intention. Together, these design choices enhance the real-world relevance of the findings and offer stronger behavioral support for the effects of destination revitalization policy.

## Study 4

Study 4 was conducted to examine whether temporal distance moderates the effect of destination revitalization policy type on tourists' behavioral responses. To gain external validity and achieve behaviorally based results, the present paper used a field experiment on real tourists and the admission of a scenic spot on a coupon of discounted travel ticket as an approximation of actual world travel behavioral intention.

### Research procedure and samples

Study 4 adopted a 2 (policy type: rigid vs. flexible) × 2 (temporal distance: short vs. long) between-subjects experimental design. Data were collected from 28 January 2024 to 16 February 2024 in the core scenic area of Zhangjiajie. The study protocol was reviewed and approved by the Ethics Committee of the School of Tourism and Urban Rural Planning, Jishou University.

Tourists visiting the destination during the data collection period were approached on-site by trained research assistants and invited to participate in the study. Participation was entirely voluntary. Before the experiment began, all potential participants were informed of the academic purpose of the study, the anonymous nature of the data collection, and their right to withdraw at any time without penalty. Verbal informed consent was obtained prior to participation.

A total of 420 tourists completed the experiment (66.7% male; 51.4% aged 36–45). Participants were randomly assigned to one of the four experimental conditions. To capture behavioral intention in a realistic manner, a scenic spot admission discount coupon valued at RMB 100 was offered, and participants' travel behavioral intention was operationalized as their decision to accept or decline the coupon.

### Research instruments

Policy type was manipulated using the same video-based stimulus materials developed and validated in Study 3, representing rigid versus flexible destination revitalization policies. The two videos were matched in length, presentation style, and informational structure to control for extraneous variation.

Temporal distance was manipulated by explicitly specifying the timing of the hypothetical trip in the experimental scenario. Participants in the short temporal distance condition were instructed to imagine visiting the destination in the following week, whereas those in the long temporal distance condition were asked to imagine the trip occurring 12 months later, following established operationalizations in prior research (e.g., Sano et al. [30]).

Following the watching of the allocated video and the reading of the temporal framing, the subjects stated the willingness to accept the discount coupon of scenic spot admission. This dichotomous option was used as the behavioral variable of the purpose of traveling. The nearly final part of the questionnaire was used to gather demographic data.

**Results.** *Manipulation checks.* The power analysis, with G*Power 3.1, before the hypothesis testing showed that the minimum sample size was 111 participants in a two-way ANCOVA ($f=0.40$, $\alpha=0.05$, $power=0.80$, four groups) [71]. The size of the final sample was beyond this value and thus sufficient statistical power. Independent-samples $t$-tests confirmed the effectiveness of the experimental manipulations. Participants clearly distinguished between rigid and flexible policies ($M_{rigid}=5.31$, $SD=1.41$; $M_{flexible}=2.15$, $SD=0.73$; $t=29.00$, $p<0.001$), as well as between short and long temporal distance conditions ($M_{short}=2.23$, $SD=0.97$; $M_{long}=5.99$, $SD=0.72$; $t=45.26$, $p<0.001$).

*Interaction effects of policy type and temporal distance.* A 2×2 ANCOVA was conducted with policy type and temporal distance as independent variables, monthly income and education as covariates, and tourists' social judgments and behavioral responses as dependent variables. The results revealed significant interaction effects between policy type and temporal distance (see Figs 4 and 5).

Under short temporal distance, flexible policies generated significantly higher perceived warmth ($M=5.70$, $SD=0.69$) than rigid policies ($M=4.05$, $SD=1.45$; $F(1, 416) = 35.58$, $p<0.001$, $\eta^2=0.08$). Consistently, binary logistic regression analysis showed that participants exposed to flexible policies were more likely to accept the discount coupon (74.80%) than those exposed to rigid policies (63.80%) ($B=1.09$, $Wald=22.91$, $p<0.001$).

In contrast, under long temporal distance, rigid policies elicited significantly higher perceived competence ($M=5.30$, $SD=1.12$) than flexible policies ($M=3.95$, $SD=1.52$; $F(1, 416) = 29.35$, $p<0.001$, $\eta^2=0.07$). Similarly, participants in the rigid policy condition exhibited a higher coupon acceptance rate (81.00%) than those in the flexible condition (70.50%) ($B=0.80$, $Wald=13.69$, $p<0.001$). These results provide empirical support for the moderating role of temporal distance.

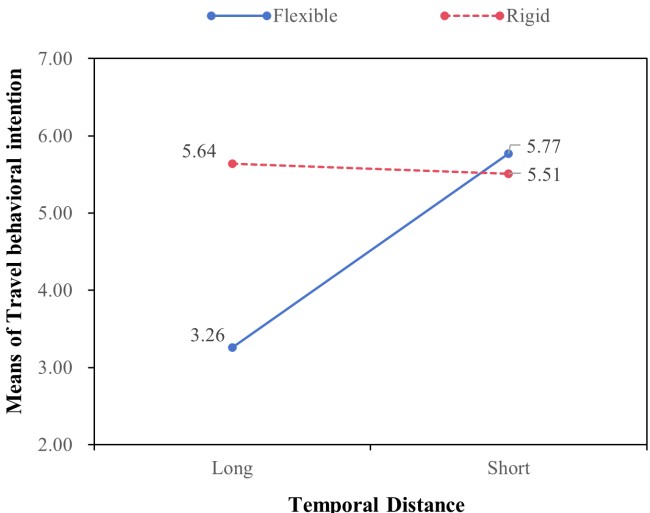

**Fig 4. Moderating effect of temporal distance between types of destination revitalization policy and travel behavioral intention (Study 4).** Source: Author's own work.

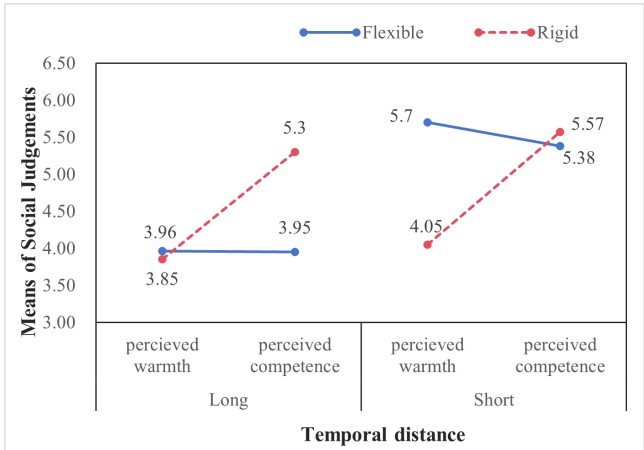

**Fig 5. Moderating effect of temporal distance between types of social judgements and travel behavioral intention (Study 4).** Source: Author's own work.

*Moderated mediation analysis.* To further examine the proposed moderated mediation effects, PROCESS Model 7 with 5,000 bootstrap samples was employed. The results showed that the indirect effect of perceived warmth was significant under short temporal distance ($\beta = -0.47$, $SE = 0.08$, 95% $CI$ [−0.64, −0.31]) but not under long temporal distance ($\beta = -0.03$, $SE = 0.06$, 95% $CI$ [−0.14, 0.09]). The index of moderated mediation was significant ($\beta = 0.43$, $SE = 0.11$, 95% $CI$ [0.24, 0.66]).

Conversely, the indirect effect of perceived competence was significant under long temporal distance ($\beta = 0.61$, $SE = 0.12$, 95% $CI$ [0.40, 0.86]) but not under short temporal distance ($\beta = 0.09$, $SE = 0.05$, 95% $CI$ [−0.01, 0.18]). The corresponding index of moderated mediation was also significant ($\beta = 0.52$, $SE = 0.13$, 95% $CI$ [0.30, 0.79]), providing further support for the proposed moderated mediation hypotheses.

**Control variable and alternative explanation.** Monthly income and education level were controlled for to account for potential demographic influences on travel-related decision-making. A 2×2 ANCOVA was conducted with policy type and temporal distance as independent variables, and monthly income and education level as covariates. The interaction effects between policy type and temporal distance remained significant for perceived warmth ($F(1, 414) = 37.53$, $p < 0.001$, $\eta^2 = 0.08$), perceived competence ($F(1, 414) = 27.90$, $p < 0.001$, $\eta^2 = 0.06$), and travel behavioral intention ($F(1, 414) = 277.34$, $p < 0.001$, $\eta^2 = 0.40$).

## Conclusions and implications

### Overall discussion

This study adopts a multi method design to test the effectiveness of destination revitalization policies. This pilot study conducted in-depth interviews with 64 tourists to determine different social judgments related to rigid and flexible policies. Study 2 conducted an online scenario based experiment on 270 potential tourists to determine the main policy effects. Study 3 further validated the mediating effect of perceived warmth and ability using a large-scale experimental sample of 265 participants. Based on these findings, Study 4 conducted field experiments on 420 on-site tourists, demonstrating that time distance moderates the impact of policies on behavioral intentions.

### Theoretical implications

This study makes three primary theoretical contributions by clarifying how destination revitalization policies operate as psychological signals, identifying the underlying social judgment mechanisms, and specifying the temporal conditions under which these mechanisms are most influential.

First, the available literature has mainly focused on revitalization policies as a tool of an institution to post-crisis recovery [11,12], to achieve competitiveness [36,37], or to have governance restructured [72–74]. Although knowledgeable, this line of studies has actually accounted except in small part, on the fact that policy effectiveness is attributable to structural or functional form, given little consideration as to how tourists think and feel about policy signals. Conceptualizing the revitalization polices as a symbolic, governing, signal and empirically differentiating between rigid and flexible policy styles [15,16], the paper proves that two policy styles of more or less similar objectives can have different behavioral effects since they trigger different social meanings. The discovery enhances the current policy frameworks by providing a behavioral and perceptions explainable finding, and thus, the research that is a destination governance and the tourist decision-making theory converge together.

Second, in pre-SCM studies, judgment about warmth and competence has mostly been induced when social groups [75], social groups [76], or organizational actors [20,77] are involved as readily recognizable social targets. On the contrary, the formation of these basic social judgments in situations where evaluative targets are in the form of impersonal but authoritative public policy is theoretically poorly articulated. The SCM applications in tourism have also focused on comparatively steady destination images instead of the dynamic interventions of policy. This research article demonstrates that warmth and competence judgments are systematically used by tourists in order to decode revitalization policies and that flexible policies induce warmth perceptions and rigid induce competence perceptions. Noteworthy, these dimensions serve as independent mediating variables between the policy design and the travel behavioral intention. This study takes SCM out of that domain by exposing how tourists anthropomorphize the styles of governance and analyze them using the key dimensions of essential social judgment and clarifying the explanatory effects it has on institutions and its practices in policy settings.

Third, research into policy evaluation has also focused, mainly, on policy content, instruments, or outcome indicators of governance [78,79], and typically, policy effects are assumed to be constant between the different decision stages. Policies judgments in the context of tourism have also been concerned with instantaneous attitudes or aggregate behavioral reactions but there has been little theoretical consideration of how time constraint of decision making affects the inherent interpretation of policy cues. Using the temporal construal theory, this research proves that the temporal distance conditionally stimulates the manner in which tourists assess governance styles implicated in revitalization policies. Temporally adjacent travel decisions lead tourists to use more affective, experience-related cues related to warmth; temporally distant travel decisions lead to the increasing significance of abstract and capability related decision making cues. This research contributes to a more dynamic conception of policy appraisal in the tourism industry by demonstrating that effectiveness of governance styles is not only a factor of policy formulation, but also temporal orientations of tourists.

## Practical implications

This research shows that flexible revitalization policies were better in temporally close decision making situations since they mainly trigger a sense of warmth and immediacy. The case of Zibo is such an example, with people-focused and highly dynamic responses, including a tolerant regulation of street vendors and responsive services to the visitors, initiating a massive rise in the number of visitors in this span of two months in early 2023, attracting over 5 million visitors. In line with our results, policies that imply warmth are proven to be an effective way to spur the demand on short-term travel, but their effects were challenging to maintain as the news of the significance of this policy started to lose its popularity in the public.

In comparison, the example of rigid tourism governance in Zhangjiajie depicts how rigid revitalization policies have their strategic position in the development of the long-term destination. The destination gained considerable competence, safety, and institutional reliability perceptions by keeping tourist complaints down (by over 60%) through long-term enforcement of the regulations, market standardization and protection of visitors, as well as, keeping the satisfaction high

                                                                                                

levels. As postulated by our findings, the competence-signaling policies have a stronger effect in decision situations that are more temporally remote, with tourists depriving themselves to more abstract measures of the quality of governance.

The flexible and rigid revitalization policies, however, play a complementary strategic role in the implementation process and do not represent a simple dichotomous decision. One can use flexible and warmth-focused policies to help new destinations with less brand recognition gain market share as quickly as possible and to help smaller-known destinations activate demand on a short-term basis.

Compared to that, in the mature destinations or a destination which places more emphasis on long-term sustainability, competence-oriented policies which are rigid offer a stronger base. In the end, both need to rely on efficient destination governance; their strict combination must be established as a base of the long-term planning and institutional responsibility whereas the loose policies may be selectively used as short-term tools to react to the market environment and provoke the immediate interest of tourists.

### Research limitations and future research directions

This research has various limitations that are subject to additional research. First, the entire sample data were gathered in China, where the cultural aspects could have affected the responsiveness of the respondents to the nature of revitalization policies. In the future, the samples used in our study should encompass different cultural settings to determine the generalizability of our results. Second, even though the conditions associated with this study were founded on real-life experiences and provided a behavioral test of the intentions to travel, the findings represent the subjective assessment of the participants according to the available materials. These results can be confirmed by future research based on big data analysis or case studies since more objective results would be obtained. Lastly, the current research identified temporal distance as the moderator between revitalization policies and the effects in the social judgments and travel intentions. Other possible moderators, though, which were not studied included destination reputation [25], perceived familiarity [80], and destination attractiveness [81]. Future studies can be conducted on the interaction of these factors with revitalization policies to determine the effects on travel behavior in various situations.

### Conclusion

Based on the stereotype content model, this paper contributes to the systematic investigation of the influencing role of destination revitalization policy types on the travel behavioral intentions of tourists. Across multiple studies, results indicate that, in all cases, flexible revitalization policy sounds greater than strict policy in creation of travel intentions. This effect operates through distinct social judgment pathways: flexible policies primarily enhance perceived warmth, whereas rigid policies strengthen perceived competence. Besides, temporal distance is a boundary condition that is critical. When travel decisions are temporally proximal, tourists rely more on warmth-based judgments, amplifying the positive impact of flexible policies. In contrast, under greater temporal distance, competence-based judgments become more salient, increasing the effectiveness of rigid policies. By integrating social judgment mechanisms with temporal construal, this research advances understanding of how governance-related policy signals influence tourist decision-making. The results offer actionable implications for destination management organizations in designing context-sensitive revitalization strategies that align with tourists' psychological evaluation processes.

### Supporting information

**S1 Fig. Rigid policy condition stimuli of Studies 1a, 2, 5.**
(TIF)

**S2 Fig. Flexible policy condition stimuli of Studies 1a, 2, 5.**
(TIF)

**S3 Fig. The influence of group type of revitalization policy on tourists' travel behavioral intention (Study 1a).**
(TIF)

**S4 Fig. Moderating effect of temporal distance between types of destination revitalization policy and travel behavioral intention (Study 5).**
(TIF)

**S5 Fig. Moderating effect of temporal distance between types of social judgements and travel behavioral intention (Study 5).**
(TIF)

**S1 File. Appendix.** This appendix contains summary information of Studies 1, 2, 3, 4 and 5 (Appendix A), interview outline for pilot study (Appendix B), experimental stimuli of destination revitalization policy (Appendix C), manipulation check questions and measurement items (Appendix D), summary tables of sample statistics across of Studies 1, 2, 3, 4 and 5 (Appendix E), and the data analysis of Study 1a, 1b, and 5 (Appendix F).
(DOCX)

## Author contributions

**Conceptualization:** Wen Qin, Juan Su.

**Funding acquisition:** Juan Su.

**Investigation:** Weilong Li, Yanli Peng.

**Methodology:** Wen Qin, Juan Su.

**Project administration:** Juan Su.

**Software:** Weilong Li.

**Validation:** Wen Qin.

**Writing – original draft:** Weilong Li, Yanli Peng.

**Writing – review & editing:** Weilong Li.

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
