## [Decision Letter · Decision Letter 0]

8 Dec 2025

PONE-D-25-40512Rigid versus flexible: The effect of destination revitalization policy on tourists' travel behavioral intentionPLOS One

Dear Dr. Su,

Thank you for submitting your manuscript to PLOS ONE. After careful consideration, we feel that it has merit but does not fully meet PLOS ONE’s publication criteria as it currently stands. Therefore, we invite you to submit a revised version of the manuscript that addresses the points raised during the review process.

Authors are recommended to make a more clear presentation and deep explanation.

We look forward to receiving your revised manuscript.

Kind regards,

Bifeng Zhu

Academic Editor

PLOS One

Journal Requirements:

“This study was supported by the National Natural Science Foundation of China (No. 72362018; 42261042), General Project of Hunan Education Department, China (No. 21C0387), and Key Project of Research project on teaching reform in ordinary higher edu-cation institutions in Hunan Province, China (No. HNJG-20230688 ).”

4. We note that your Data Availability Statement is currently as follows: “All relevant data are within the manuscript and its Supporting Information files.”

Reviewers' comments:

Reviewer's Responses to Questions

**Comments to the Author**

1. Is the manuscript technically sound, and do the data support the conclusions?

Reviewer #1: Yes

Reviewer #2: Yes

2. Has the statistical analysis been performed appropriately and rigorously? 

Reviewer #1: Yes

Reviewer #2: Yes

3. Have the authors made all data underlying the findings in their manuscript fully available?

Reviewer #1: Yes

Reviewer #2: Yes

4. Is the manuscript presented in an intelligible fashion and written in standard English?

Reviewer #1: Yes

Reviewer #2: Yes

5. Review Comments to the Author

Reviewer #1: Thank you for allowing me to review the paper entitled “Rigid versus flexible: The effect of destination revitalization policy on tourists' travel behavioral intention" Here are some suggestions to improve the paper:

1. Originality: Does the paper contain new and significant information adequate to justify publication?

1. The current form of the abstract is not well written. Author should highlight the novelty and main aim of the study. Make sure it commences with a succinct statement of the primary aim of the study, highlighting the novelty it brings to the field. This should be followed by a concise description of the research methodology employed, including the analysis techniques utilized and details regarding the sample collection process. Finally, it should conclude with a brief summary of the key findings and the implications derived from them. It is essential to emphasize the unique contributions of this study. Only highlight the most significant findings and implications to maintain focus.

2. The introduction lacks a clear structure and focus. Please follow this suggested structure: (1) Summarize the existing issue together with its importance. (2) Explain how researchers should address the discovered areas that need further investigation. (3) The research gap must clearly relate to the existing issue. (4) The research benefits from critiquing these aspects that previous academic studies avoid. (5) Review relevant past studies. (6) Draw a contrast between this study and all related past research. (7) Clearly state the research objectives. (8) The paper details how it introduces beneficial concepts to academic literature. A properly organized introduction creates better clarity by establishing solid groundwork for the overall paper.

3. The issue is not explicitly explained and support with recent citations.

4. The authors fail to define their research gap with clarity and supporting evidence. The authors need to present a precise identification of the unaddressed areas in past research. All necessary arguments should be presented to validate the immediate need for addressing this gap. Provide a clear reasoning that demonstrates the study’s connection to contemporary conditions.

2. Relationship to Literature: Does the paper demonstrate an adequate understanding of the relevant literature in the field and cite an appropriate range of literature sources? Is any significant work ignored?

5. Should have a standalone section for underpinning theory. This is to explain how the theory supporting the proposed paths and model. To enhance the clarity and coherence of the research, I recommend the authors thoroughly explain how or any theory support the current study and each path. By elaborating on the theory's key concepts and demonstrating its relevance to the research topic, the authors can establish a stronger connection between the theoretical foundation and the study's objectives.

6. Author does not explain how the theory supporting the mediation paths and moderation paths. Please include this justification in the underpinning theory section.

7. A proper flow for literature review as follows: (1) underpinning theory, (2) hypotheses development and (3) proposed research model.

3. Methodology: Is the paper's argument built on an appropriate base of theory, concepts, or other ideas? Has the research or equivalent intellectual work on which the paper is based been well designed? Are the methods employed appropriate?

8. The research methodology is not well written. Should have two sub-sections: (1) Research procedure and samples and (2) research instruments.

9. Why there is a need to slit the study into five studies?

10. How do you test cause and effect relationship?

11. The authors should discuss the generalizability and representativeness of their sample in relation to the target population. Authors need to clearly explain how the chosen sample is intended to be representative of and reflective of the larger population. Any strategies employed to ensure a diverse and inclusive sample should also be highlighted. This will increase the credibility of the research findings and help readers understand the extent to which the results can be generalized to the broader population.

12. It is essential to provide a clear explanation of a sampling technique used in the study, along with the rationale for its selection. Why these techniques? The authors should describe how they utilized this sampling technique to select respondents for the survey, ensuring generalizability and representativeness towards the targeted population. By doing so, readers can better understand the methodological approach and the potential limitations associated with the sample selection.

13. Any selection criteria?

14. Any control variable included in this study? Why not as it could be one of the confounding variables?

15. The procedure of data collection needs to be elaborated further. The authors should explain in more detail how they collected the data, how they approached the respondents, and how they identified participants for the survey study. This explanation should be reasonable and logical, avoiding exaggerations and providing a clear account of the steps taken in the data collection process.

16. Did you translate the questionnaire to local language? If yes, please explain how it was done.

17. The research instrument section is not well written. Should refer to quality paper published in high impact journal on how to report the research instruments.

4. Results: Are results presented clearly and analysed appropriately? Do the conclusions adequately tie together the other elements of the paper?

18. The reporting is confusing.

19. For common method bias, I suggest using a full collinearity test.

20. The analysis results reporting is not clear. Author should report the preliminary analysis result (common method bias test, descriptive analysis results, and respondents profile characteristics), measurement model analysis (reliability, convergent validity, discriminant validity), and structural model analysis (path result)

5. Practicality and/or Research implications: Does the paper identify clearly any implications for practice and/or further research? Are these implications consistent with the findings and conclusions of the

21. The structure of the conclusion part should be as follows: (1) Discussion, (2) Theoretical Implications, (3) Practical Implications, (4) Limitations and Future Research Recommendations, and (5) Conclusion.

22. Should have a overall discussion section.

23. For theoretical implication, it is too shallow. This section should discuss the implications of the study's findings and how they contribute to the existing theoretical knowledge. Summarize the key findings and their relevance to the existing theoretical frameworks or models. Analyze how the findings align with or challenge current theoretical perspectives and concepts related to all the key concepts of this study. Discuss any theoretical insights or advancements that the study provides and highlight how the findings contribute to a deeper understanding of the research area.

24. For practical implications, it is not well written. The authors should provide valuable insights based on current practices and policies, supported by evidence from their research. To strengthen the practical implications, it is crucial to reference specific findings, data, or examples that demonstrate the validity and reliability of the recommendations. By incorporating this approach, the authors can offer concrete and actionable suggestions that have a solid grounding in their research findings.

25. A conclusion section with not more than 150 words is required.

26. Proofread the manuscript, Clarity is required, and cite more recent relevant studies. Please also check citation format and referencing style.

Reviewer #2: I don’t see it as mandatory because it is already a complete piece of work, but I think that the methodological section could be strengthened if some key aspects related to the internal validity of the experimental designs were described in more detail:

(1) Extraneous or confounding variables: It would be helpful to identify which variables could influence the results and to explain clearly how they are controlled (for example, randomization, counterbalancing, standardized instructions, or environmental control). In the last studies, this would be especially important because the designs are becoming more complex.

(2) Controls used to address threats to internal validity: such as history, maturation, selection, and experimenter effects. It would be useful to indicate which procedures were used to reduce these threats.

6. PLOS authors have the option to publish the peer review history of their article (what does this mean?). If published, this will include your full peer review and any attached files.

Reviewer #1: No

Reviewer #2: **Yes:** Cristian O. Beltran-Oicata

To ensure your figures meet our technical requirements, please review our figure guidelines: s://journals.plos.org/plosone/s/figures

You may also use PLOS’s free figure tool, NAAS, to help you prepare publication quality figures: s://journals.plos.org/plosone/s/figures#loc-tools-for-figure-preparation.

---

## [Author Response · Author response to Decision Letter 1]

22 Jan 2026

Dear Dr. Zhu and the Reviewers,

We sincerely thank you and the editorial team for the careful handling of our manuscript and for the time and effort devoted to its evaluation. We are also grateful to the two reviewers for their constructive, detailed, and insightful comments, which have been extremely helpful in improving the quality and clarity of our work.

In response to the editor’s guidance and all reviewer comments, we have conducted a thorough and careful revision of the manuscript. Given the scope and depth of the revisions, we have provided a detailed, point-by-point response in the attached file entitled “Response to Reviewers.docx.” In this document, we address each comment individually and clearly indicate where the corresponding changes can be found in the revised manuscript, with specific line numbers referring to the “Revised Manuscript with Track Changes.docx.”

Overall, the revision focused on four main aspects. First, we substantially improved the clarity and coherence of the manuscript by reorganizing the logical structure of the theoretical framework, methods, results, discussion, and implications. Second, we strengthened the methodological reporting by providing more explicit descriptions of experimental procedures, internal validity considerations, and ethical approvals. Third, we deepened the theoretical discussion by rearticulating the core theoretical contributions and clarifying how the findings address specific gaps in the existing literature. Finally, we revised the practical implications to ensure that they are clearly grounded in the empirical findings and offer concrete, evidence-based guidance for policy and practice.

We hope these revisions have significantly improved the rigor, transparency, and overall contribution of the manuscript. We sincerely appreciate the editor’s and reviewers’ valuable feedback and hope that the revised version now meets the journal’s expectations. We would be grateful for your further consideration of our manuscript.

Thank you very much for your consideration.

Kind regards,

Juan Su

---

## [Decision Letter · Decision Letter 1]

26 Feb 2026

PONE-D-25-40512R1Rigid versus flexible: The effect of destination revitalization policy on tourists' travel behavioral intentionPLOS One

Dear Dr. Su,

Thank you for submitting your manuscript to PLOS ONE. After careful consideration, we feel that it has merit but does not fully meet PLOS ONE’s publication criteria as it currently stands. Therefore, we invite you to submit a revised version of the manuscript that addresses the points raised during the review process.

**Please check the reference and figures.**

We look forward to receiving your revised manuscript.

Kind regards,

Bifeng Zhu

Academic Editor

PLOS One

**Journal Requirements:**

Reviewers' comments:

Reviewer's Responses to Questions

**Comments to the Author**

1. If the authors have adequately addressed your comments raised in a previous round of review and you feel that this manuscript is now acceptable for publication, you may indicate that here to bypass the “Comments to the Author” section, enter your conflict of interest statement in the “Confidential to Editor” section, and submit your "Accept" recommendation.

Reviewer #1: All comments have been addressed

Reviewer #2: All comments have been addressed

2. Is the manuscript technically sound, and do the data support the conclusions?

Reviewer #1: Yes

Reviewer #2: Yes

3. Has the statistical analysis been performed appropriately and rigorously? 

Reviewer #1: Yes

Reviewer #2: Yes

4. Have the authors made all data underlying the findings in their manuscript fully available?

Reviewer #1: Yes

Reviewer #2: Yes

5. Is the manuscript presented in an intelligible fashion and written in standard English?

Reviewer #1: Yes

Reviewer #2: Yes

6. Review Comments to the Author

**Reviewer #1:** I am satisfied with the revised manuscript. All my comments are addressed sufficiently. Thus, I have no further comments.

**Reviewer #2:** Dear author

Congratulations on the rigor of your research

Please, in terms of style, correct the reference error on line 203 and label the figures as "Author's own work".

7. PLOS authors have the option to publish the peer review history of their article (what does this mean?). If published, this will include your full peer review and any attached files.

Reviewer #1: No

Reviewer #2: **Yes:** Cristian Oswaldo Beltrán-Oicatá

To ensure your figures meet our technical requirements, please review our figure guidelines: s://journals.plos.org/plosone/s/figures

You may also use PLOS’s free figure tool, NAAS, to help you prepare publication quality figures: s://journals.plos.org/plosone/s/figures#loc-tools-for-figure-preparation.

---

## [Author Response · Author response to Decision Letter 2]

13 Apr 2026

Dear Editor and reviewers ,

We sincerely thank you and the reviewers for your careful evaluation of our manuscript and for your valuable comments.

In this revision, we have carefully addressed all comments raised by the editorial office and the reviewers. Specifically, we have reviewed the reference list to ensure completeness and accuracy, corrected formatting and citation issues, and revised the identified reference error. In addition, we have updated all figure captions to include “Author’s own work,” as suggested.

All changes have been incorporated into the revised manuscript. For clarity, the revised content is highlighted in red in the file “Revised Manuscript with Track Changes.docx.” A detailed, point-by-point response to each comment is provided in the attached file “Response to Reviewers.docx.”

We greatly appreciate the constructive feedback and the positive evaluation of our work. Thank you again for your time and consideration.

Kind regards,

Juan Su

On behalf of the authors

---

## [Editor Report · Decision Letter 2]

14 Apr 2026

Rigid versus flexible: The effect of destination revitalization policy on tourists' travel behavioral intention

PONE-D-25-40512R2

Dear Dr. Su,

We’re pleased to inform you that your manuscript has been judged scientifically suitable for publication and will be formally accepted for publication once it meets all outstanding technical requirements.

Kind regards,

Bifeng Zhu

Academic Editor

PLOS One
---

## [Editor Report · Acceptance letter]

PONE-D-25-40512R2

PLOS One

Dear Dr. Su,

I'm pleased to inform you that your manuscript has been deemed suitable for publication in PLOS One. Congratulations! Your manuscript is now being handed over to our production team.

Kind regards,

on behalf of

Dr. Bifeng Zhu

Academic Editor

PLOS One